# Association between Inflammatory Potential of Diet and Bone-Mineral Density in Korean Postmenopausal Women: Data from Fourth and Fifth Korea National Health and Nutrition Examination Surveys

**DOI:** 10.3390/nu11040885

**Published:** 2019-04-19

**Authors:** Woori Na, Susan Park, Nitin Shivappa, James R. Hébert, Mi Kyung Kim, Cheongmin Sohn

**Affiliations:** 1Department of Food and Nutrition, Wonkwang University, 460 Iksandaero, Iksan, Jeonbuk 54538, Korea; nawoori6@gmail.com (W.N.); tntks19@naver.com (S.P.); 2Cancer Prevention and Control Program, University of South Carolina, Columbia, SC 29208, USA; SHIVAPPA@mailbox.sc.edu (N.S.); JHEBERT@mailbox.sc.edu (J.R.H.); 3Department of Epidemiology and Biostatistics, Arnold School of Public Health, University of South Carolina, Columbia, SC 29208, USA; 4Department of Nutrition, Connecting Health Innovations LLC, Columbia, SC 29201, USA; 5Division of Cancer Epidemiology and Prevention, National Cancer Center, 323, Ilsan-ro, Ilsandong-gu, Goyang-si, Gyeonggi-do 10408, Korea

**Keywords:** dietary inflammatory index, bone-mineral density, menopause, osteoporosis

## Abstract

Post menopause is considered a critical period for bone-mass loss. Impaired bone metabolism during this phase can increase the risk of fractures in old age. Inflammation is a risk factor for bone health, and diet is a potential source of inflammation. However, few studies have examined the association between the dietary inflammatory index (DII^®^) and bone-mineral density (BMD) in postmenopausal women in Korea. The objective of this study was to determine, by means of a Korean cross-sectional investigation, whether higher DII scores are associated with decreased BMD in postmenopausal women. To that end, the raw data from the fourth and fifth Korea National Health and Nutrition Examination Surveys (KNHANES, 2009–2011) for 2778 postmenopausal women aged over 50 years were analyzed. The subjects’ BMD were measured by dual-energy x-ray absorptiometry, and their DII^®^ scores were calculated from a single 24-h dietary recall. Further, the participants were classified into three groups according to DII^®^ score. Women with more pro-inflammatory diets (i.e., those in the highest tertile of DII^®^) had significantly lower BMD in the femoral neck as compared with women in the lowest tertile (*p* for trend <0.05) after adjustment for age, body-mass index (BMI), household income, education status, smoking habits, physical activity, total calcium intake, female-hormone use, age at menopause, and blood vitamin D levels. Multiple logistic regression analyses revealed that the odds ratio (OR) of total femur osteopenia/osteoporosis was higher in women in the highest tertile of DII^®^ than in those in the lowest (OR 1.27, 95% CI 1.00-1.62, *p* for trend < 0.05). This study established that more pro-inflammatory diets might be associated with lower BMD in postmenopausal Korean women.

## 1. Introduction

Korean society is aging rapidly. In 2017, people aged 65 and above constituted 14% of the total population [1]. By 2026, Korea is expected to become a “super-aged society” with an elderly population of more than 20% [1]. Osteoporosis is a typical senile disorder in which bone strength decreases with decreasing bone-mineral density (BMD) and microstructural abnormality, thereby increasing the risk of fracture [2,3]. It is known that age and female-hormone-related status are the main etiologies of osteoporosis. Other risk factors for osteoporosis are weight, education level, nutrition, and physical activity [4]. Since it is impossible to alter genetic factors, managing adjustable factors is required to maintain bone strength and BMD, which eventually helps to reduce the risks of osteoporosis and fracture [5]. Due to the fact that osteoporotic fractures in the elderly impose a great health care burden on them [6,7], it is important to understand the risk factors of osteoporosis and manage them, not to mention diagnose and treat them early. Inflammation is closely related to bone remodeling, which might be involved in the pathogenesis of postmenopausal osteoporosis. Pro-inflammatory cytokines such as Intereukin-1 (IL-1), Interleukin-6 (IL-6), and tumor necrosis factor-alpha (TNF-α) are important regulators of bone resorption [8,9]. It has been reported that deficiency of estrogen causes increased IL-1 production by cellular monocytes, and that TNF-α levels increase in women who experience osteoporotic fractures [10]. Certainly, the pro-inflammatory mediators play an important role in estrogen deficiency-associated bone loss in postmenopausal women. 

Reportedly, inflammation is modified by dietary factors; in this light, a focus on overall diet quality rather than on single nutrients or foods might be more appropriate to efforts to explain inflammatory mechanism [11,12].

Shivappa et al. conducted research to evaluate inflammation levels through the analysis of various dietary components, finally developing the dietary inflammatory index (DII^®^), which assesses the overall inflammatory quality of an individual’s diet [13]. The pro-inflammatory dietary factors to be considered are carbohydrates, saturated fatty acids, proteins, and others, and the anti-inflammatory factors are micronutrients such as low unsaturated fatty acids, flavonoids, vitamin A, vitamin C, vitamin D, vitamin E, and selenium, among others. The DII^®^ has been validated with various inflammatory markers in various countries (PMID: 30096775, PMID: 29675557, PMID: 28448077, PMID: 25639781, PMID: 29936717) [14,15,16,17,18], including in Korea [19]. Previous studies have suggested a significant association between DII^®^ and bone health. In small cross-sectional studies on postmenopausal Iranian women, it has been reported that women with higher DII^®^ scores were more likely to have below-median BMD in the lumbar spine [20]. In another longitudinal study based on Women’s Health Initiative data, women with the least inflammatory dietary patterns suffered greater lower-hip BMD loss [21]. In Kim et al.’s study, which analyzed data on osteoporosis and diagnoses based on patients’ self-reports, an increase in osteoporosis risk with an increase in DII^®^ among postmenopausal women was found [22]. However, a cross-sectional analysis of DII^®^ and BMD in young adult subjects showed no such correlation [23]. In any case, given that the risk of osteoporosis is significantly increased in postmenopausal women, its incidence is expected to increase among Korea’s aging population. Therefore, it is important to evaluate the relationship between DII^®^ and bone health in postmenopausal women vulnerable to bone loss.

Osteoporosis is a debilitating disease that causes high rates of fragility fractures and high morbidity. Although the rates of hip fracture in the United States, France, Spain, and Australia are decreasing, incidence rates are increasing in Korea [24]. Even so, only one study thus far has investigated the relationship between DII^®^ and BMD in postmenopausal Korean women [25]. 

The aim of the present study was to investigate the association between DII^®^ and BMD as well as the risk of osteopenia/osteoporosis in a representative sample of postmenopausal Korean women.

## 2. Materials and Methods

### 2.1. Study Population

The study population was selected form the fourth and fifth Korea National Health and Nutrition Examination Surveys (KNHANES) conducted by the Korean Ministry of Health (Sejong City, Korea and Welfare from 2009–2011. These were nationwide representative studies that employed a stratified sampling design for selection of households. Details on the study design and methods are available elsewhere [26]. The study population for this analysis included only postmenopausal women who had completed nutrition surveys, physical examinations, and BMD measurements by dual-energy X-ray absorptiometry. We excluded a total of 12,490 women: those of premenopausal status (*n* = 10,783), those aged less than 50 years (*n* = 296), those suffering or who had suffered from serious chronic disease (e.g., stroke, myocardial infarction, angina pectoris, liver cirrhosis, renal failure, and cancer) (*n* = 283), those who had missing dietary data (*n* = 710), those who had a caloric intake of less than 500.0 kcal/day or one that was equal to or more than 4000.0 kcal/day (*n* = 66), and those with missing DXA data (*n* = 268). Thus, data from 2778 subjects were used in the final analysis (Figure 1). The present study was ethics approved by the Clinical Test Deliberation Commission of the Institutional Review Board (No. WKIRB-201801-SB-007) (Wonkwang University, Iksan City, Korea).

### 2.2. Analysis of DII^®^

The dietary intakes of the participants were assessed by the 24-h dietary recall method. The DII^®^ had been developed based on a systematic literature review of the relationships between food parameters and blood inflammatory indices. The parameters of the DII^®^ applied in this study included six foods (garlic, ginger, onion, turmeric, green tea/black tea, and pepper) and 35 nutrients (alcohol, caffeine, carbohydrate, protein, cholesterol, energy, fat, dietary fiber, vitamin B_12_, vitamin B_6_, beta-carotene, folic acid, iron, magnesium, monounsaturated fatty acids, polyunsaturated fatty acids, niacin, omega-3 fatty acids, omega-6 fatty acids, thiamine, riboflavin, saturated fatty acids, trans fats, selenium, vitamin A, vitamin C, vitamin D, vitamin E, zinc, anthocyanidins, flavan-3-ols, flavanones, flavones, flavonols, and isoflavones). To calculate the DII^®^ of each subject, the dietary intake was standardized using the mean and standard deviation of a global composite dataset from 11 countries (USA, Australia, Bahrain, Denmark, India, Japan, New Zealand, Thailand, Korea, Mexico, and Britain) and converted to a percentile score. The DII^®^ score of each person was calculated by multiplying the percentile score and the respective inflammatory effect score of all food parameters, and then summing them. The content data on the nutrients used in this study were obtained from the Rural Development Agency Materials [27], the Computer Aided Nutritional Analysis Program (2011 for Korean Nutrition Society, Version 4.0), and the United States Department of Agriculture database [28]. A low DII^®^ is indicative of an anti-inflammatory diet, and a high DII^®^ of a pro-inflammatory diet.

### 2.3. Bone-Mineral Density (BMD)

The BMDs of the lumbar spine [L1-L4], total femur, and femoral neck were measured by DXA scan (DISCOVERY-W fan-beam densitometer, Hologic Inc., Santa Clara, CA, USA). According to the measured BMD and WHO criteria, the groups with T scores < −1.0 and > −2.5 were defined as osteopenic, and the groups with T scores < −2.5 were defined as osteoporotic [29].

### 2.4. Covariates

Demographic information, including age, education level (primary school, middle school, high school graduation, college graduate or higher), and household income (low, moderately low, moderately high, high), was collected via self-administered questionnaires. Lifestyle information, including smoking status (current, never, and past), and physical activity (active, non-active) was investigated. Physically active was defined as follows: engage in vigorous physical activity for more than 20 min at least three times a week, or light/moderate physical activity for more than 30 min at least five times a week. Details regarding age at menopause, calcium supplementation, and female-hormone use also were obtained. Blood 25-hydroxyvitamin D (25(OH) Vitamin D, ng/mL) concentrations were calibrated by radioimmunoassay (RIA).

### 2.5. Statistical Analysis

We used a complex sample analysis to reflect the sampling method of KNHANES. The participants were divided into tertiles according to DII^®^ scores, and an analysis of variance (ANOVA) and Chi-square test were performed for each variable, including age, body-mass index (BMI), education level, income level, physical activity, smoking status, blood 25(OH) Vitamin D concentration, and classification of BMD (lumbar spine, total femur, and femoral neck). The characteristics of the subjects were presented as means and standard deviations, or percentages. To determine statistical differences in the average BMD according to each DII^®^ group, an analysis of covariance (ANCOVA) that was adjusted for covariates was used for each subject’s lumbar spine, total femur, and femoral neck. We reassessed the relationship between bone health and DII^®^ by multivariate logistic regression analysis that defined the dependent variable as less than −1 in T score. The adjusted variables were as follows: age (continuous), BMI (classified), household income (classified), smoking habits (classified), physical activity (classified), calcium supplementation (classified), use of female hormone (classified), and Blood 25-hydroxyvitamin D (continuous). Statistical significance was considered as a *p* < 0.05, and all of the statistical analyses were completed with the Predictive Analysis SoftWare (PASW) package (version 24.0: SPSS Inc., Chicago, IL, USA).

## 3. Results

### 3.1. Characteristics of Study Participants

Participants’ characteristics according to the DII^®^ are presented in Table 1. Across increasing DII^®^ scores, mean age increased from 61.28 to 65.81 years DII^®^ (*p* < 0.001). The proportions of subjects with higher education and household income decreased with increasing DII^®^ score (*p* < 0.001). The subjects with higher DII^®^ scores were more likely to have lower BMI scores (*p* < 0.05). There were health-behavior differences, such as smoking habits, for different DII^®^ scores. There were significant differences in calcium intake and use of calcium supplementation according to tertiles of DII^®^ (*p* < 0.001).

### 3.2. BMD According to DII^®^

The comparison of BMD according to DII^®^ intake is shown in Table 2. As the DII^®^ score increased, the BMD of femoral neck and total femur turned out to be lower after adjusting for bone formation-related factors such as age, physical activity, postmenopausal period, and blood 25-hydroxyvitamin D (*p* for trend <0.05).

### 3.3. Risk of Osteopenia/Osteoporosis According to DII^®^

The analysis results of the crude and adjusted odds ratios (OR) for risk of osteopenia/osteoporosis according to the three tertiles of DII^®^ are listed in Table 3. The multivariate logistic regression models indicated a significantly increased OR of osteopenia/osteoporosis in the total femur (T2; OR 1.39, 95% CI 1.09–1.76, T3; OR 1.27, 95% CI 1.00–1.62, *p* trend < 0.05) and femoral neck (T2; OR 1.18, 95% CI 0.93–1.49, T3; OR 1.43, 95% CI 1.10–1.86, *p* trend < 0.01) for the subjects in the highest tertiles of DII^®^ after adjusting for potential confounders.

## 4. Discussion

The association between osteoporosis and aging is a well-known epidemiological phenomenon that can cause serious health problems among the elderly. In our representative sample of postmenopausal Korean women, we found that increased DII^®^ was associated with lower BMD and increased risk of osteoporosis in femur bone only. 

Several studies have investigated the relationship between DII^®^ scores and bone health. One, with a nationwide sample of Americans aged between 40 and early 50s, demonstrated that a lower mean value of BMD in total femur and femoral neck is associated with higher DII^®^ in both men and women [30]. However, in a study on postmenopausal Iranian women, subjects with higher DII^®^ scores were not associated with BMD in the femoral neck [20]. On the other hand, in another study with postmenopausal white women, it was found that the BMD of the lumbar spine was rather higher in the group with higher DII^®^ and that the proportion of older participants was lower in the group with higher DII^®^ [21]. In contrast to this study, our present investigation illustrated that across increasing tertiles of DII^®^, the subjects’ mean age increased while the BMD of femur bone decreased, but not in the lumbar spine. In other words, our results indicated the importance of following a less-inflammatory diet to maintain bone health in postmenopausal women. The results for the relationship between BMD and DII^®^ varied with ethnicity, age, and gender as well as with the site of BMD measurement. 

Since BMD is a representative risk factor with the highest predictive capability for fracture, it is widely used as a basic indicator for management of general bone health [29,31,32]. Among Korean women, the peak BMD of the lumbar spine and total hip was reached at different times at different skeletal sites: 30–39 years of age at the lumbar spine and 40–49 years of age at the total hip [33]. Since BMD changes are displayed differently for each bone part according to postmenopausal period, caution is required for the interpretation of results. Further, since in most postmenopausal women bone loss occurs first in the lumbar vertebrae, which have more trabecular bone, the lumbar-spine BMD T-value is lower than the femur neck BMD T-value [34]. Thus, lumbar-spine osteoporosis in Korean women starts in the early 50 s, while femoral osteoporosis starts to appear in the mid−50 s and rapidly increases in the 60 s. Additionally, the proportion of osteoporosis in the femur has been found to increase with age (50 s: 14%, 60 s: 18%, 70 s: 26%, 80 s: 32%) [35]. This means that the relative risk of osteoporosis varies by bone site and age. Considering that the mean age of the subjects in our study was over 60 years, it is worth speculating that bones with relatively high risk of osteoporosis are more sensitive to inflammatory diets.

Previous studies have demonstrated associations between the inflammatory cytokines and bone health. IL-1 and IL-6 have uncoupled bone remodeling by enhancing bone resorption and suppressing bone formation [9]. Higher high-sensitivity C-reactive protein levels have been associated with lower trabecular bone score and bone-quality index [36]. Inflammatory cytokines have mediated bone loss directly by stimulating osteoclast formation and maturation or indirectly by promotion of ligand-RANKL release [37]. Various studies have presented evidence that dietary difference could have an effect on inflammatory cytokines secretion [38]. A normal human diet consists of both pro-inflammatory and anti-inflammatory food parameters. The DII^®^, which was developed to measure the actual impact of dietary inflammatory potential [13], has been found to be associated with several inflammatory cytokines including C-reactive protein, IL-6, and homocysteine [17,39]. The present study found that higher DII^®^ scores were associated with higher osteoporosis risk. We can speculate that dietary intake influences the inflammatory cytokines levels, which could modulate bone remodeling by uncoupling bone resorption and formation.

Osteoporosis is a chronic disease that is affected by various nutrients and food intakes; meanwhile, BMD loss begins to appear more conspicuously post menopause. It is important to control nutritional intake to prevent osteoporosis after middle age [40]. However, according to the results of a previous comparative study on dietary quality in Korean women, the mean adequacy ratio (MAR) of postmenopausal women was lower than that of premenopausal women [41]. Furthermore, considering the increasing rate of fractures in Korean women [24], the implementation of nutritional intervention programs focusing on total dietary quality and anti-inflammatory regimes should be further emphasized to prevent osteoporosis and public health burdens in women after middle age. 

The limitations of this study are as follows. First, causal inference on the association between DII^®^ and BMD is limited because this was a cross-sectional study. Second, even though BMD represents an accumulation of years of bone changes reflective of long-term dietary exposure, the DII^®^ was calculated using 24-h recall data. Third, the questionnaire-derived data utilized in this study might not accurately reflect the subject’s usual dietary intake. Despite these limitations, this study has several strengths. First, it revealed the relationship between DII^®^ and BMD using nationally representative data. Second, our data were adjusted for several demographic factors, lifestyles, and dietary factors as confounders. 

In conclusion, we found that higher DII^®^ scores were associated with increased likelihood of bone-health abnormalities in postmenopausal women. This knowledge could have implications for dietary advice provided to those at risk of osteopenia/osteoporosis. 

## Figures and Tables

**Figure 1 nutrients-11-00885-f001:**
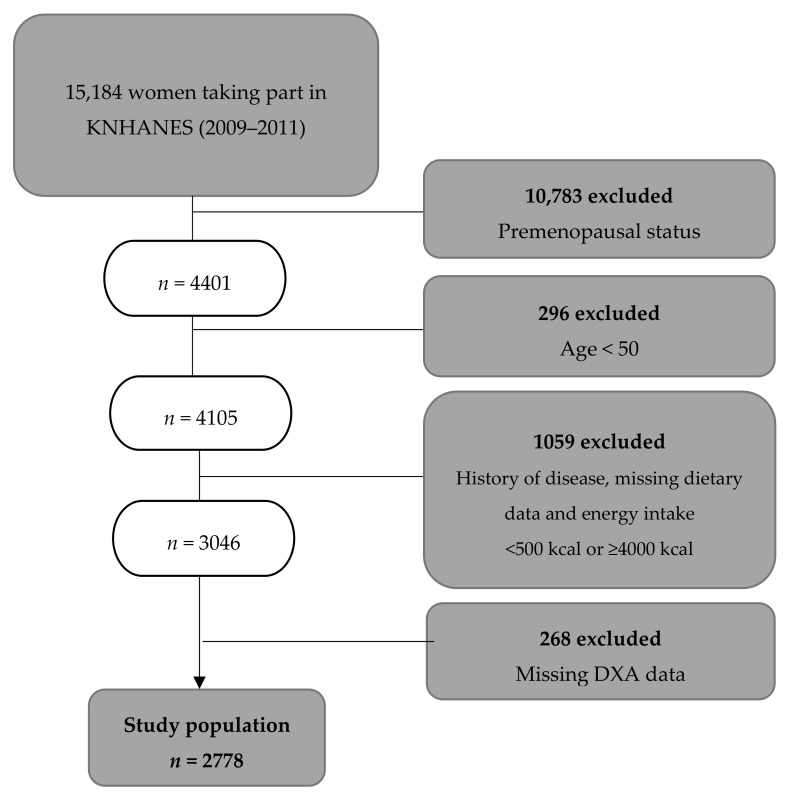
Flow chart.

**Table 1 nutrients-11-00885-t001:** Characteristics of participants according to dietary inflammatory index (DII^®^) score.

Variable	T1 ^a^ (*n* = 926)	T2 (*n* = 926)	T3 (*n* = 926)	*p* Value
Age (years) ^b^	61.28 ± 7.73	63.88 ± 8.76	65.81 ± 8.84	<0.001
Body-mass index (kg/m^2^)	24.29 ± 3.07	24.24 ± 3.23	23.95 ± 3.11	0.046
Education				<0.001
≤Primary	474 (51.3)	617 (66.8)	689 (74.6)	
Middle school	163 (17.6)	150 (16.3)	107 (11.6)	
High school	215 (23.2)	115 (12.5)	99 (10.7)	
≥College graduate	73 (7.9)	41 (4.4)	29 (3.1)	
Household income				<0.001
Low	247 (26.9)	350 (38.2)	399 (43.5)	
Moderately low	225 (24.5)	249 (27.2)	239 (26.1)	
Moderately high	216 (23.5)	174 (19.0)	145 (15.8)	
High	230 (25.1)	144 (15.7)	134 (14.6)	
Physical Activity				0.68
Physically active	123 (13.3)	93 (10.0)	94 (10.1)	
Physically non-active	803 (86.7)	833 (90.0)	832 (89.9)	
Smoking				0.001
None	876 (94.7)	852 (92.0)	834 (90.2)	
Ex-smoker	6 (0.6)	4 (0.4)	8 (0.9)	
Current smoker	43 (4.6)	70 (6.7)	82 (9.0)	
Vitamin D level (ng/mL)	17.92 ± 6.34	18.50 ± 6.94	17.81 ± 6.86	<0.001
Postmenopausal period (years)	12.23 ± 9.26	14.65 ± 10.34	17.09 ± 10.76	<0.001
Female-hormone supplementation	205 (22.1)	146 (15.8)	117 (12.6)	<0.001
Calcium supplementation	183 (19.8)	160 (17.3)	121 (13.1)	<0.001
Calcium intake (mg) ^c^	561.19 ± 12.69	421.19 ± 11.37	300.85 ± 12.51	<0.001
DII^®^ Score	−0.88 ± 1.03	1.98 ± 0.63	4.12 ± 0.70	<0.001

DII^®^: Dietary inflammatory index. ^a^ Tertile DII^®^ score range: T1 (−5.15 to 0.84), T2 (0.85 to 3.04), and T3 (3.05 to 6.53). ^b^ Values are mean ± SD or percentage. ^c^ Data were adjusted for energy intake (kcal/day) by GLM analysis.

**Table 2 nutrients-11-00885-t002:** Bone-mineral density (BMD) by DII^®^ Score.

Variable	T1 ^a^ (*n* = 926)	T2 (*n* = 926)	T3 (*n* = 926)	*p* Value	*p* for Trend
Total femur BMD					
Crude ^b^	0.799 ± 0.004	0.771 ± 0.004	0.751 ± 0.004	<0.001	<0.001
Adjusted 1 ^c^	0.782 ± 0.003	0.772 ± 0.003	0.767 ± 0.003	0.004	<0.001
Adjusted 2 ^d^	0.782 ± 0.003	0.775 ± 0.003	0.771 ± 0.003	0.07	0.025
Femoral neck BMD					
Crude	0.648 ± 0.004	0.619 ± 0.004	0.599 ± 0.004	<0.001	<0.001
Adjusted 1	0.631 ± 0.003	0.62 ± 0.003	0.615 ± 0.003	0.001	<0.001
Adjusted 2	0.631 ± 0.003	0.623 ± 0.003	0.618 ± 0.003	0.01	0.004
Lumbar-spine BMD					
Crude	0.820 ± 0.005	0.794 ± 0.005	0.777 ± 0.005	<0.001	<0.001
Adjusted 1	0.804 ± 0.004	0.795 ± 0.004	0.792 ± 0.004	0.08	<0.001
Adjusted 2	0.802 ± 0.004	0.797 ± 0.004	0.797 ± 0.004	0.6	0.36
Whole-body total BMD					
Crude	1.025 ± 0.004	1.001 ± 0.004	0.991 ± 0.004	<0.001	<0.001
Adjusted 1	1.012 ± 0.004	1.002 ± 0.004	1.003 ± 0.004	0.11	0.002
Adjusted 2	1.010 ± 0.004	1.001 ± 0.004	1.006 ± 0.004	0.25	0.39

BMD: Bone-mineral density (g/cm^2^). ^a^ Tertile DII^®^ score range: T1 (−5.15 to 0.84), T2 (0.85 to 3.04), and T3 (3.05 to 6.53). ^b^ Estimated mean (values are mean ± SE). ^c^ Adjusted for age, BMI. ^d^ Adjusted for age, BMI, household income, smoking habits, physical activity, female-hormone use, postmenopausal period, and Vitamin D.

**Table 3 nutrients-11-00885-t003:** Multivariate adjusted odds ratios (OR, 95% confidence intervals) for osteopenia + osteoporosis by tertiles of DII^®^ score.

Variable	T1 ^a^ (*n* = 926)		T2 (*n* = 926)		T3 (*n* = 926)		*p* for Trend
	Case	OR(CI)	Case	OR(CI)	Case	OR(CI)	
Total femur osteopenia + osteoporosis							
Crude ^b^	252	1 (Ref)	373	1.80 (1.48, 2.19)	415	3.03 (2.49, 3.70)	<0.001
Adjusted 1 ^c^		1		1.42 (1.14, 1.78)		1.35 (1.08, 1.69)	0.007
Adjusted 2 ^d^		1		1.39 (1.09, 1.76)		1.27 (1.00, 1.62)	0.04
Femoral neck osteopenia + osteoporosis							
Crude	648	1	720	1.50 (1.21, 1.84)	778	2.25 (1.80, 2.82)	<0.001
Adjusted 1		1		1.23 (0.97, 1.55)		1.53 (1.19, 1.96)	0.11
Adjusted 2		1		1.18 (0.93, 1.49)		1.43 (1.10, 1.86)	0.007
Lumbar-spine osteopenia + osteoporosis							
Crude	663	1	712	1.32 (1.07, 1.62)	751	1.70 (1.37, 2.12)	<0.001
Adjusted 1		1		1.11 (0.89, 1.38)		1.21 (0.96, 1.53)	0.001
Adjusted 2		1		1.03 (0.82, 1.29)		1.11 (0.87, 1.49)	0.42

DII^®^: Dietary inflammatory index, ref: reference. ^a^ Tertile DII^®^ score range: T1 (−5.15 to 0.84), T2 (0.85 to 3.04), and T3 (3.05 to 6.53). ^b^ Multivariate logistic regression models presented with odds ratio (OR) and 95% confidence interval (CI). ^c^ Adjusted for age, BMI. ^d^ Adjusted for age, BMI, household income, smoking habits, physical activity, calcium intake, female-hormone use, postmenopausal period, and Vitamin D.

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
