# Peer review of "Association between Inflammatory Potential of Diet and Bone-Mineral Density in Korean Postmenopausal Women: Data from Fourth and Fifth Korea National Health and Nutrition Examination Surveys"

_nutrients, 2019, doi:10.3390/nu11040885_

Round 1

Reviewer 1 Report

This is a very interesting report on the association between the Inflammatory Potential of Diet and Bone-Mineral Density in Korean  Postmenopausal Women. The manuscript is very well written, the hypothesis is clearly stated and the results support their conclusion, namely, that " more pro-inflammatory diets might be associated with lower BMD in postmenopausal Korean women". These results  have relevant clinical implications relative to the importance of following a less-inflammatory diet to maintain bone health in postmenopausal women. follows. The authors acknowledge some limitations (namely the cross sectional nature of this study  and that the DII® index was calculated using 24-hour data. However, the strengths (the large number of subjects studied and the fact that the data was adjusted for several demographics, lifestyles, and dietary factors) out weight these limitations. 

Author Response

Thank you for your review of this study.

Reviewer 2 Report

Overall, this manuscript is a clear and easy to follow. The introduction provides a good and clear background. The methodology is clear as well.  The authors used correct statistical analysis to analyze their data. In line 162, physical activity is not matching with the results of table 1. I did not see the physical activity is different between three groups.  Maybe the authors did not show the p value.  The discussion relates to the results.  

Author Response

Overall, this manuscript is a clear and easy to follow. The introduction provides a good and clear background. The methodology is clear as well.  The authors used correct statistical analysis to analyze their data. In line 162, physical activity is not matching with the results of table 1. I did not see the physical activity is different between three groups.  Maybe the authors did not show the p value.  The discussion relates to the results.  

Response: Thank you for your review. There was no significant difference in physical activity according to tertile of DII. Therefore, we modified the sentence accordingly. 

Reviewer 3 Report

Minor revision:

Lines 72-76 in Introduction is similar to lines 202-206 in Discussion.

Lines 235-236 in Discussion should be re-written

References should be written uniformly. The authors sometimes write  DOI: 10...... and sometimes  DOI: https.//doi.org/10.......

Author Response

Point 1 : Lines 72-76 in Introduction is similar to lines 202-206 in Discussion.

Response : We have modified the sentences in lines 202-206, which are similarly written.

Point 2: Lines 235-236 in Discussion should be re-written

Response : We have modified the sentence in lines 235-236.

Point 3 : References should be written uniformly. The authors sometimes write DOI: 10...... and sometimes DOI: https.//doi.org/10.......

Response : We were have modified the DOI forms in the reference.

Reviewer 4 Report

It will be really better if the author can replace the first 10 (except the 1 and 8) references with new updated last 10 year references or add new references in those places along with the present one.

 The author mentioned about female hormone supplement, which female hormone supplement was that?

Author Response

It will be really better if the author can replace the first 10 (except the 1 and 8) references with new updated last 10 year references or add new references in those places along with the present one.

Answer: We have replaced references 2, 3, 4, 6, 7, and 9 with the latest updated papers.

The author mentioned about female hormone supplement, which female hormone supplement was that?

Answer: Women's hormones used the results of an examination question: " Have you ever take a female hormone or had an injection since the menopause?" There was no distinction between the types of female hormones in the question.
